# An Adaptive Trust Evaluation Model for Detecting Abnormal Nodes in Underwater Acoustic Sensor Networks

**DOI:** 10.3390/s24092880

**Published:** 2024-04-30

**Authors:** Changtao Liu, Jun Ye, Fanglin An, Weili Jiang

**Affiliations:** 1School of Cyberspace Security, Hainan University, Haikou 570228, China; liuchangtao@hainanu.edu.cn (C.L.); anfanglin@hainanu.edu.cn (F.A.); jiangweili@hainanu.edu.cn (W.J.); 2Key Laboratory of Internet Information Retrieval of Hainan Province, Haikou 570228, China

**Keywords:** underwater acoustic sensor networks, adaptive trust model, fuzzy comprehensive evaluation, fuzzy closeness, anomaly detection

## Abstract

Underwater acoustic sensor networks have a wide range of applications in both civil and military fields, but the complex and changing underwater environment makes them vulnerable to multiple security threats. Trust mechanisms are effective ways to enhance network security and reliability. In order to improve the accuracy of trust evaluation and the detection rate of abnormal nodes, this paper proposes an adaptive trust evaluation model based on fuzzy logic. This model adopts a variable weight fuzzy comprehensive evaluation algorithm to dynamically adjust the weights of three direct trust indicators to ensure the accuracy of direct trust evaluation. Then, it uses fuzzy closeness to eliminate unreliable recommendation trust and adjusts the weight of recommendation trust through deviation to improve the accuracy of indirect trust. The simulation results show that the model can effectively improve the accuracy of trust evaluation and the detection rate of abnormal nodes. Especially when the link quality is unstable, the success rate of detecting abnormal nodes in this model is improved by more than 10% compared with the existing trust model.

## 1. Introduction

In recent years, the application scope of underwater acoustic sensor networks (UASNs) has continuously expanded, ranging from marine scientific research to underwater resource detection and further to military reconnaissance and communication, demonstrating significant potential and value [1]. These networks leverage the characteristics of underwater sound wave propagation, constructing a widely covered and interconnected sensing system. However, with increasing applications, UASNs face various challenges [2], including the complexity of the underwater environment, noise interference, and signal attenuation, significantly impacting network stability and reliability [3]. Therefore, overcoming these challenges and enhancing the performance of underwater acoustic sensor networks is of profound significance for advancing underwater communication and detection technologies.

Currently, conventional security technologies such as identity authentication and security encryption can effectively resist external intruder attacks. However, once attackers successfully infiltrate the system, these technologies often struggle to cope [4]. In recent years, trust mechanisms have shown outstanding performance in detecting internal attacks. Trust, as a social concept, essentially reflects the subjective trust level in specific entity behavior, involving a complex cognitive process. In UASNs, each node evaluates its trustworthiness based on the past behavior of other nodes. By constructing a trust model, we can effectively identify and eliminate abnormal nodes in the network, thereby enhancing the overall security performance of the network [5].

In UASNs, due to the sparse distribution and frequent movement of sensor nodes, most nodes cannot communicate directly [6]. Additionally, malicious nodes in the network may launch attacks against other nodes while releasing benign signals to assessment nodes, disrupting trust evaluations. Therefore, relying solely on direct trust to determine node trustworthiness is insufficient. To avoid over-reliance on the subjective judgment of assessment nodes in trust calculations, the introduction of third-party node recommendation information is considered [7]. However, a core issue lies in the difficulty of ensuring the quality of recommendation information. Due to the instability of link quality, even minor biases in recommended trust may result in severe biases in the assessment of overall node performance [8]. This increases the difficulty and uncertainty of trust evaluation, requiring the exploration of more reliable and effective methods to address these challenges.

Despite significant progress made in enhancing the security of UASNs through trust models proposed in previous research and the adoption of effective filtering strategies to deal with unreliable recommendations, several challenges remain unresolved. Currently, many trust models often employ a simplistic threshold-based approach to filter seemingly unreliable recommendations. However, this method lacks effective means to deter malicious nodes, allowing them to disseminate misleading information within the network. Moreover, these models typically use a simple weighted average method to compute the final recommendation trust value for nodes, which may not accurately reflect the trustworthiness of the nodes. Furthermore, node trust values are not static but dynamically adjust over time and with changes in the network environment. This dynamic variation not only reflects the real-time nature of node behavior but also enhances the flexibility of trust evaluation.

To filter out anomalous nodes in the network and enhance its security and reliability, this paper proposes an adaptive trust evaluation model (ATEM) that dynamically assesses the trustworthiness of nodes. The study’s main contributions are as follows:(1)Considering the harsh underwater communication environment, it is necessary to evaluate the link quality between nodes first during trust assessment. Based on this evaluation, it is determined whether trust assessment should be conducted in the current cycle, reducing the impact of link quality on the accuracy of trust evaluation.(2)An adaptive fuzzy comprehensive evaluation algorithm based on variable weighting is proposed to comprehensively consider the multidimensional trust indicators of the target node. By dynamically adjusting the weights of each trust indicator, this algorithm highlights deficiencies in certain trust factors of the target node, thereby enhancing the accuracy of the trust evaluation model.(3)Fuzzy closeness theory is employed to calculate indirect trust, filtering out recommended nodes with significant deviation from the center. Through this process, nodes with a large deviation are removed, and weights are allocated based on the degree of deviation, thereby improving the reliability of indirect trust.

The remaining sections of this work are structured as follows. Section 2 summarizes relevant work on trust models. Section 3 describes the network concept and operating mechanism of the ATEM trust model. In Section 4, we conducted a simulation to assess the ATEM’s performance. Section 5 presents a summary of the article.

## 2. Related Work

In recent years, researchers have continuously proposed various trust models suitable for UASNs based on previous studies. However, due to the complexity of the underwater environment, existing trust models still face various challenges. In this section, we briefly review the existing trust models from three specific aspects: link quality assessment, direct trust calculation, and recommendation trust filtering.

The instability of link quality often leads to a decline in the interaction effect between sensor nodes, making it necessary to consider the impact of link quality on trust evaluation. Su et al. designed a novel trust model that specifically addresses the issue of underwater communication link instability. By introducing a link quality assessment method, they accurately measured its impact on trust calculations. Compared to traditional models, this trust model more comprehensively considers the impact of poor communication link quality on the trust values of sensor nodes, especially the negative effects on the trust values of normal nodes [9]. Han et al. proposed an attack-resistant trust model based on multidimensional trust metrics. During trust calculations, this model comprehensively considers various indicators of link trust, such as link quality and link capacity. It also delves into the analysis of the unreliability of communication channels and the mobility of the underwater environment. Through these considerations, the model significantly improves the accuracy of trust evaluation [10]. Su et al. presented a unique trust management mechanism that introduces a trust redemption process to address the issue of normal nodes being misclassified as malicious nodes. This process considers the historical performance of nodes and environmental factors, thereby reducing the likelihood of misjudging normal nodes [11].

In direct trust calculation, existing trust models comprehensively consider various trust metrics for evaluation based on practical application requirements. However, the weights of trust metrics in most trust models often rely on subjective experience and are set as fixed values, lacking objectivity and accuracy. Jiang et al. proposed an efficient distributed trust model suitable for wireless sensor networks. When calculating direct trust, this model considers communication trust, energy trust, and data trust and subjectively sets the weights of these three trust metrics. Then, the direct trust value is obtained through weighted summation [12]. Wu et al. presented a trust model based on link quality indicators. When calculating direct trust, it considers communication trust, energy trust, and data trust and discusses the weights of these three trust metrics in different situations, improving the accuracy of direct trust [13]. Ye et al. introduced an efficient dynamic trust evaluation model for wireless sensor networks. This model achieves accurate and efficient trust evaluation by dynamically adjusting the weights of direct trust and indirect trust, as well as updating mechanism parameters. To achieve accurate trust evaluation, the model considers multiple trust metrics, including communication trust, data trust, and energy trust, and incorporates penalty factors and adjustment functions to calculate direct trust. Additionally, a sliding window update mechanism based on the induced ordered weighted averaging operator is proposed, which can dynamically adjust parameters and the number of interaction history windows according to the actual needs of the network, enabling dynamic updates of direct trust values [14].

In order to more accurately evaluate the trust of nodes, recommendation trust is often introduced for comprehensive evaluation. This inevitably leads to the hidden danger of false recommendation trust. This misleading and false information may seriously interfere with the evaluation of the trust value. Zhang et al. proposed a recommendation management trust mechanism based on collaborative filtering and variable weight fuzzy algorithms. This mechanism combines communication, data, and energy trust to calculate node trust values and obtains overall recommendation trust values through a collaborative filtering algorithm. It can effectively filter unreliable recommendations and improve the recognition rate and stability of the trust model [8]. Du et al. presented a defective recommendation filtering scheme. The core of this scheme lies in the preliminary screening of recommendation information using the median of cluster head nodes, followed by updating the recommendation trust values of nodes through a collaborative filtering algorithm. This approach effectively avoids errors that may be introduced due to the improper setting of empirical thresholds and successfully filters out some false recommendation values, thereby improving the accuracy of recommendation trust [6]. Anwar et al. adopted the Bayesian estimation method for calculating recommendation trust values. This method can fully utilize existing evidence information and accurately estimate recommendation trust values based on event probabilities. However, the Bayesian estimation method relies on the setting of prior probabilities, which means that relevant probability values need to be stored in a database when calculating new probabilities. Although this data storage mechanism helps to improve calculation accuracy, it also inevitably increases storage space usage [15]. Sun et al. introduced a recommendation information filtering method that focuses on using divergence detection degrees to deeply analyze the reliability of recommendation values. By comparing the differences between recommendation values and detection degrees, they can effectively identify and filter out recommendation information that may involve unfair recommendations or collusion attacks. When the difference between the recommendation value and the detection degree exceeds a certain threshold, the system rejects the recommendation value, ensuring the accuracy and reliability of the recommendation data [16]. Pang et al. proposed an optimization method that combines a fuzzy trust model with an artificial bee colony algorithm for calculating indirect trust. Through a precise detection mechanism, once a recommender node with dishonest behavior is identified, these unreliable nodes are immediately added to a blacklist. However, it is worth noting that the model has limitations in distinguishing between erroneous recommendations and dishonest recommendations [7]. Alnasser presented a novel approach for calculating recommendation trust values, with the core lying in an adaptive weighting mechanism [17]. This weighting mechanism dynamically adjusts based on the number of positive and negative recommendations, aiming to more accurately reflect the reliability and effectiveness of recommendation information. Adewuyi et al. introduced an innovative trust function designed to quantify the degree of acceptance of recommendations and allocate weights accordingly to obtain the final recommendation trust value [18].

Based on the aforementioned trust models and addressing various existing issues, this paper proposes an adaptive trust evaluation model for detecting anomalous nodes in underwater acoustic sensor networks. This model enables the effective filtering of unreliable recommendation information and identification of anomalous nodes through a more scientific evaluation of node behavior and recommendation quality. Compared with traditional trust assessment methods, this model has a higher recognition rate and stability, especially in the face of typical attack scenarios, where it can show a better performance.

## 3. Network Model and Link Quality Assessment

In this section, the network model and link quality assessment are described. First, it describes the network model, then introduces the structure of the trust evaluation model, and finally details the process of link quality evaluation.

### 3.1. Network Model

This paper focuses on UASNs uniformly distributed in three-dimensional space, where the sensor nodes possess identical capabilities. Each sensor node has a unique ID for identification purposes. Communication between nodes is only possible when they enter each other’s communication range, and for non-directly adjacent nodes, they rely on the forwarding function of other nodes to exchange data information, assuming that sensor nodes can accurately determine their underwater coordinates and, through interaction with surrounding nodes, acquire information about the remaining energy status of adjacent nodes. Additionally, they have storage capabilities to dynamically update the neighbor list for maintaining the stability of network connections.

Figure 1 depicts the workflow of the ATEM, which includes the five main steps listed below: (1) link quality assessment; (2) direct trust computation; (3) indirect trust computation; (4) composite trust computation; and (5) trust update.

In this paper’s suggested trust architecture, the evaluating node will check the connection quality status with the target node at the start of each time cycle. If the connection quality does not fulfill the criterion, the trust evaluation for the current cycle will be halted, and the evaluating node will wait until the next cycle to reassess. Once the link quality matches the requirements, node A will generate three trust indicators and use the fuzzy comprehensive assessment procedure to compute the direct trust value. If the direct trust value does not fulfill the criterion, node A will obtain recommendation information for node B from all of their close nearby nodes. ATEM will then filter based on the closeness of target node B’s direct trust values to each recommending node and assign weights depending on the departure of the suggested trust values from the central value, resulting in the final indirect trust value. Finally, using a balancing weight factor, ATEM will balance the weights of direct and indirect trust to obtain the composite trust value for target node B.

### 3.2. Link Quality Assessment

In view of the inherent characteristics of the underwater environment, such as significant delay, limited bandwidth, signal attenuation phenomena, and the instability of sound speed and channel communication quality fluctuating with the environment, it is generally challenging to maintain consistent connection quality. Poor network quality frequently reduces the interaction impact between sensor nodes, resulting in frequent data packet loss and retransmissions. This not only increases excessive energy consumption, but it also has a detrimental influence on the standard trust evaluation between nodes. 

The underwater environment presents unique characteristics compared to terrestrial or aerial environments, such as higher signal attenuation rates and increased obstacles and interference to signal transmission. Due to these characteristics, traditional metrics of link quality such as RSSI and LQI may be less reliable in underwater environments. However, the packet reception ratio (PRR) is a metric that directly reflects the successful transmission of data packets over a communication link, unaffected by signal attenuation or interference. Therefore, using the PRR for evaluating link quality in underwater environments can provide more reliable results. However, merely deriving the packet reception ratio (PRR) through a straightforward calculation of the ratio between the number of correctly received packets and the total transmitted packets may not effectively discern the quality of the link. Conversely, we lean towards placing trust in links that have recently demonstrated a higher volume of successful data receptions, as we perceive these links to exhibit a superior performance. Therefore, we can assign different weights to each successfully received data packet in the link to more accurately differentiate link quality. For a given sequence of data packet reception status records, denoted as s(i), where successful and unsuccessful reception statuses are represented by 1 and 0, respectively, greater weight should be assigned to records closer to the current time. Thus, by normalization, we can obtain the weight value for each data packet reception status record in the link as $w(i)=\frac{2i}{n(n+1)}$, where i refers to the index in the list of packet reception status records, and n is the total number of reception status records. Then, the link status can be calculated by the following formula:(1)$$L=\sum_{i}^{n}w(i)\times s(i)$$When L is less than the threshold ε, it means that the current link status is unstable; otherwise, the link quality is considered to be good.

## 4. Trust Calculation

This section mainly explores trust computation, encompassing four core components: direct trust, indirect trust, comprehensive trust, and trust update. In the calculation of direct trust, we first compute the trust values for three direct trust indicators separately. Subsequently, we adopt a variable weight fuzzy comprehensive evaluation algorithm to dynamically adjust the weights of each indicator, and finally fuse them to obtain the direct trust value. In terms of indirect trust calculation, we eliminate recommendation trusts with large deviations through fuzzy closeness, and then dynamically assign weights to each recommendation trust based on the deviation, ultimately fusing them to derive the indirect trust value. Furthermore, we obtain the comprehensive trust value by integrating direct and indirect trusts. Lastly, we achieve dynamic trust updates by combining trust values from historical and current cycles.

### 4.1. Direct Trust

#### 4.1.1. Trust Indicators Generation

There are various forms of internal attacks in UASNs, and to defend against different types of attacks, the selection of trust indicators should reflect these attack behaviors. While the more trust indicators selected, the more accurate the calculated trust value will be, the limited energy and computing capabilities of underwater wireless sensors prevent the consideration of all factors. Thus, there exists a balance between the selection of trust factors and the capabilities of the sensors themselves. Considering both attack behaviors and the accuracy of trust values, this paper selects the following three trust indicators.

(1)Communication Trust

The communication behavior of nodes, especially the success and failure of communication during the time period T, has a direct impact on their communication trust within that period. In UASNs, the instability of communication channels is a significant characteristic, often affected by various factors such as environmental interference. This reliance solely on previous communication conditions to detect node communication behavior introduces considerable uncertainty. To address the impact of this uncertainty on trust assessment, we can employ subjective logic theory to compute the communication trust of nodes, with the following formula:(2)$$T_{com}=\frac{2b+u}{2}$$
where $b=\frac{s}{s+f+1}$, $u=\frac{1}{s+f+1}$. s and f indicate the number of successful and failed communications, respectively. 

(2)Data Trust

The transmission of data between adjacent regions exhibits spatiotemporal correlation, allowing neighboring nodes to receive similar data information. Consequently, tampered data by malicious nodes will exhibit noticeable disparities from normal data. Assuming the perceived data by adjacent nodes follow a normal distribution, where the mean effectively represents the majority of data points, the mean value serves as a crucial metric for assessing data similarity and reliability. Thus, data trust can be represented as follows: (3)$$T_{data}=2(0.5-\int_{\mu }^{x_{d}}f(x)dx)$$
where $x_{d}$ represents the data from the target node, and μ refers to the mean value of this data set. As the gap between $x_{d}$ and μ increases, the reliability of the data decreases accordingly.

(3)Energy Trust

In UASNs, due to limited energy, it naturally becomes a key indicator for measuring node reliability and performance. In a normal network environment, node energy consumption remains stable. However, when malicious nodes launch attacks, their node energy consumption tends to exhibit abnormal fluctuations. To appropriately measure the energy trustworthiness of nodes, we compute it using the target’s remaining energy. If the target node’s remaining energy is insufficient, it may be unable to complete its assigned responsibilities, and its energy trustworthiness value will be set to 0. Thus, the approach for calculating energy trust is the following:(4)$$T_{eng}=\left\{\begin{array}{cc} \frac{E_{res}}{E_{ini}}; & E_{res}\geq \delta \\ 0; & else \end{array}\right.$$
where $E_{ini}$ and $E_{res}$ represent the initial energy and remaining energy of the target node, respectively.

#### 4.1.2. Direct Trust Calculation

The purpose of this work is to reliably determine the direct trust value of nodes using a data fusion approach that incorporates several trust variables. Currently, weighted-averages-based trust models frequently rely on subjective experience to calculate weights, resulting in a lack of impartiality and precision. This study presents a variable-weight fuzzy comprehensive assessment method to improve trust evaluation accuracy and fairness. The basic principle behind this approach is that when the value of a trust factor is low, suggesting a possible negative influence on network performance, we raise its weight. This allows the fused direct trust value to better represent the node’s real performance in the network, especially when malicious nodes are present. The potential harm they do will be emphasized, making the final trust value more accurate and dependable. The implementation stages for this technique are as follows: (1)Establish trust factor set and direct trust evaluation set

First, the trust factor set $T=\left\{T_{com},T_{data},T_{eng}\right\}$ is established through the three trust factors selected above, and then we define three fuzzy evaluation sets $U=\left\{u_{1},u_{2},u_{3}\right\}$, where $u_{i}(i=1,2,3)$, which respectively represent “complete distrust”, “slightly trust”, and “highly trust”, reflecting the level of trust between nodes.

(2)Build membership matrix

We use membership functions to convert the values of each trust indicator into the corresponding membership degrees of the evaluation subsets. In this paper, we employ trapezoidal membership functions to obtain the membership matrix. Each row of the membership matrix E reflects the degree of membership of the corresponding direct trust factor to each evaluation subset, as shown below:(5)$$E=\left[\begin{array}{ccc} e_{11} & e_{12} & e_{13}\\ e_{21} & e_{22} & e_{23}\\ e_{31} & e_{32} & e_{33} \end{array}\right]$$
where $e_{ij}$ represents the affiliation relationship between the trust dimension factor $T_{i}$ and the fuzzy evaluation subset $u_{i}$.

(3)Confirm weights

For a given set of trust indicators $T=\left\{T_{1},T_{2},T_{3}\right\}$, the corresponding base weights $w_{mi}(i=1,2,3)$ can be obtained using the analytic hierarchy process (AHP), representing the weights when all trust indicators are at their optimal levels, satisfying $\sum_{i=1}^{3}w_{mi}=1$. Assuming trust indicator $T_{i}$ is at its worst while all other trust indicators are at their best, the weight $w_{0i}(i=1,2,3)$ can be calculated using the following equation:(6)$$w_{0i}=\frac{w_{mi}}{\min w_{mj}+\max w_{mj}}(1\leq j\leq 3)$$

We can introduce a non-increasing function $\gamma_{i}(T_{i})\in [0,1]$, where the maximum value of $T_{i}$ within its range can be represented as $\gamma_{i}(0)=\gamma_{0i}$, and the minimum value can be represented as $\gamma_{i}(1)=\gamma_{mi}$. Therefore, the weight $w_{i}(i=1,2,3)$ corresponding to $T_{i}$ can be calculated using the following formula:(7)$$w_{i}=\frac{\gamma_{i}}{\sum_{j=1}^{3}\gamma_{j}}$$

From the above equation, it can be observed that $w_{i}$ can be used as a variable weight and a non-increasing function about $T_{i}$. $\sum_{i=1}^{3}w_{i}T_{i}$ is a non-decreasing function of $T_{i}$.

(4)Fuzzy Comprehensive Evaluation Result

By utilizing the previously calculated trust weight vector W and the membership matrix E, we can compute the result vector of fuzzy synthesis evaluation according to the following formula:(8)D=W∘E
where ∘ represents the fuzzy synthesis operator, and in this paper, we employ a weighted averaging operator. Subsequently, we utilize the centroid method to defuzzify the evaluation result vector, obtaining the direct trust value within this cycle.

### 4.2. Indirect Trust

The relationship of indirect trust is illustrated in Figure 2, where node i serves as the evaluation node, node j as the target node, and nodes k and l as common neighbors of nodes i and j, respectively. When assessing the indirect trust of a target node, the fuzzy proximity theory is applied to address potential false recommendation information. This method evaluates the proximity of each neighboring node’s direct trust value to the target node and allocates weights reasonably based on the deviation between the recommended trust value and the central value. Consequently, the final indirect trust value is obtained. 

The evaluation process begins by screening the recommended trust values of common neighboring nodes, followed by weight allocation according to the deviation of these nodes’ recommended trust values. Ultimately, the indirect trust value can be obtained using the following calculation formula:(9)$$IT_{ij}(t)=\sum_{k\in S_{f}}w_{k}\times RT_{ij}^{k}(t)$$
where $S_{f}$ represents the node set after filtering the common neighbor nodes, $w_{k}$ represents the weight of any node in the neighbor node set, and $RT_{ij}^{k}(t)$ denotes the recommended trust value of neighbor node k to target node j at time t. Through this method, we can more accurately quantify the indirect trust level of the target node, thereby providing more accurate data support for overall trust evaluation.

Since trust possesses the property of transitivity in the network, when node k serves as a common neighbor, its recommended trust value $RT_{ij}^{k}(t)$ for target node j can be calculated using the following formula:(10)$$RT_{ij}^{k}(t)=DT_{ik}(t)\times DT_{kj}(t)$$

In the same period T and the same monitoring area, since the external environment of the nodes is the same, the values of the same trust factor collected by different neighbor nodes k for the target node j should be similar and tend to a certain value. If false recommendations exist in the network, their recommendation information will noticeably deviate from that of neighboring nodes. Therefore, based on the above analysis, the concept of closeness in fuzzy mathematics can be employed to calculate the degree of proximity of each common neighboring node to the target node’s direct trust value within the same period T.

In the same period T, we define the proximity $\rho_{kl}$ of common neighboring nodes k and l to the target node j’s direct trust value as:(11)$$\rho_{kl}=\frac{\min\left\{DT_{kj}(t),DT_{lj}(t)\right\}}{\max\left\{DT_{kj}(t),DT_{lj}(t)\right\}}$$

Assuming there are q neighboring nodes between the evaluating node and the target node, the proximity matrix M of common neighboring nodes to the target node can be obtained as follows:(12)$$M=\left[\begin{array}{cccc} 1 & \rho_{12} & \cdots & \rho_{1q}\\ \rho_{21} & 1 & \cdots & \rho_{2q}\\ \vdots & \vdots & \ddots & \vdots \\ \rho_{q1} & \rho_{q2} & \cdots & 1 \end{array}\right]$$

The sum of each row element in matrix M, represented as $\sum_{l=1}^{q}\rho_{kl}$, signifies the trust proximity of a common neighboring node k with other neighboring nodes. Since it is necessary to remove the proximity of neighboring nodes to themselves, the average trust closeness $\bar{\rho_{k}}$ between the common neighbor node k and other surrounding neighbor nodes is defined as follows:(13)$$\bar{\rho_{k}}=\frac{\sum_{l=1,l\neq k}^{q-1}\rho_{kl}}{q-1}$$

The above expression reflects the trust closeness of neighboring node k to target node j compared to other neighboring nodes. If $\bar{\rho_{k}}$ is relatively large, it indicates that neighboring node k is close in trust value to the surrounding neighboring nodes regarding target node j, implying that this neighboring node has higher credibility. Conversely, if $\bar{\rho_{k}}$ is relatively small, it indicates that neighboring node k deviates from the trust values of the surrounding neighboring nodes regarding target node j, suggesting that the credibility of this node is low, and it may provide false recommendation trust values to the evaluating node.

To ensure the accuracy of recommended trust, it is possible to filter out potentially malicious recommendations by comparing the deviation between the recommended trust and the central value. We define the deviation degree $d_{k}$ between the average trust closeness of a common neighbor node and other surrounding neighbor nodes and the total trust closeness as follows:(14)$$d_{k}=\left|\bar{\rho_{k}}-\rho_{0}\right|$$
where $\rho_{0}$ represents the sum of the trust proximities of all common neighboring nodes. Its calculation formula is given below:(15)$$\rho_{0}=\sum_{k=1}^{q}\sum_{l=1,l\neq k}^{q-1}\rho_{kl}$$

If the value of $d_{k}$ is larger, it reflects that the trust closeness of common neighbor node k to target node j deviates from the central value. Then, the recommended trust of neighbor node k is likely to be a malicious false recommendation, and its recommended trust value should be discarded. Therefore, in order to filter out malicious false recommendation trust, the deviation threshold value θ is defined to filter out the recommendation trust with deviation $d_{k}>\theta$. For neighbor nodes with $d_{k}\leq \theta$, stored in the set $S_{f}$, the trust value of malicious false recommendations can be filtered.

However, the method of filtering out malicious false recommendation trust values by setting the deviation threshold θ and obtaining the reliable recommendation trust node set $S_{f}$ will not eliminate all malicious recommendation trust. Since the deviation reflects the extent to which the recommended trust value of neighboring nodes deviates from the central value, for a more accurate assessment of the target node’s indirect trust, we can allocate different weights to each node in the neighbor node set $S_{f}$ based on their deviation. The larger the deviation $d_{k}$, the greater the probability that neighbor node k provides false recommendations. Therefore, in data fusion, its weight should be smaller. We define the total deviation degree D(t) of node recommendation trust in the set $S_{f}$ as follows:(16)$$D(t)=\sum_{k\in S_{f}}d_{k}(t)$$

The numerical relationship $r_{k}(t)$ between the total offset degree D(t) of node recommendation trust in the set $S_{f}$ and any offset degree $d_{k}(t)$ is as follows:(17)$$r_{k}(t)=\frac{D(t)}{d_{k}(t)}$$

Then, the weight $w_{k}$ of any node k in the neighbor node set $S_{f}$ can be calculated through normalization:(18)$$w_{k}=\frac{r_{k}(t)}{\sum_{k\in S_{f}}r_{k}(t)}$$

The final indirect trust $IT_{ij}(t)$ is calculated as follows:(19)$$IT_{ij}(t)=\sum_{k\in S_{f}}w_{k}\times RT_{ij}^{k}(t)=\frac{\sum_{k\in S_{f}}r_{k}(t)\times DT_{ik}(t)\times DT_{kj}(t)}{\sum_{k\in S_{f}}r_{k}(t)}$$

### 4.3. Comprehensive Trust

To solve the issue of constantly shifting weights for direct and indirect trust in the computation of comprehensive trust values, a balance weight factor is proposed. The computation of comprehensive trust values is often represented as shown below: (20)$$T_{ij}(t)=\varphi DT_{ij}(t)+(1-\varphi )IT_{ij}(t)$$
where φ is the balance weight factor, satisfying φ∈[0,1], and the value of φ denotes the weight of direct and suggested trust in the entire trust computation process from node i to target node j. Fixed experience values are commonly employed in literature to integrate trust. Due to the dynamic mobility of undersea nodes, which might join or leave the network at any moment, we include a balancing weight factor function, designated as φ, to guarantee that the trust evaluation model appropriately represents the actual situation.
(21)$$\varphi =f(n)=\frac{1}{2}+\frac{1}{\pi }\arctan\left(10\times \frac{n-N_{th}}{N}\right)$$
where φ dynamically changes with the variation in interaction count n, thus dynamically adjusting the weights. And N represents the maximum possible interaction count between nodes, and $N_{th}$ is a specific threshold. When the interaction count n exceeds the threshold $N_{th}$, it implies that there is sufficient interaction between the evaluating node and the target node. Therefore, we can rely more on direct trust and appropriately increase the weight of direct trust. Conversely, when the interaction count n is minimal, the target node is relatively unfamiliar compared to the evaluation result. The direct trust value cannot accurately reflect the true reliability of the target node. Hence, the evaluating node can rely more on indirect trust to assess the credibility of the target node.

### 4.4. Trust Update

Given that malevolent nodes might pose as genuine entities to boost their trustworthiness, node trust levels must be updated on a frequent basis. This method enables a more rapid reflection of the network’s current state, assuring the correctness of trust evaluation and successfully identifying and isolating potential malicious nodes in order to preserve network security and stability. Given the temporal sensitivity of trust, we present a sliding time window approach that uses trust values from several time cycles as historical trust factors to update trust values dynamically. This approach ensures that we fully consider the historical behavior of nodes when evaluating trust, thereby achieving more reliable trust assessment.
(22)$$T_{ij}(t)=\mu T_{ij}^{c}(t)+(1-\mu )T_{ij}(t-1)$$
where $T_{ij}^{c}(t)$ represents the comprehensive trust value of the current period, and $T_{ij}(t-1)$ represents the comprehensive trust value of the previous historical period. For trust value, μ represents the adaptive weight factor.

## 5. Simulation Results and Performance Analysis

In this section, we established a simulation environment using MATLAB R2020b to conduct experimental simulations on ATEM and performed comparative analysis on its performance. In the experimental setup, 100 nodes were randomly deployed within a 1000 m × 1000 m × 1000 m area, following the MCM model for mobility [19]. The communication range of each node was set to 400 m. First, our main goal was to investigate how the trust model’s performance is affected by the variable-weight fuzzy comprehensive evaluation technique. Experimental verification indicated the algorithm’s great usefulness in improving trust evaluation accuracy and dynamic flexibility. We then validated the usefulness and accuracy of the suggestion filtering method using fuzzy set progress theory. The experimental findings show that this technique may successfully filter out false suggestion information, increasing the overall reliability of trust evaluation. Finally, we evaluated the proposed ATEM against two existing trust models, ARTMM and LTrust. The comparative experimental findings support the benefits of ATEM in trust evaluation for UASNs. The specific simulation parameters are shown in Table 1.

### 5.1. Performance of ATEM

#### 5.1.1. Variable Weight and Constant Weight

We compared the variable-weight fuzzy comprehensive evaluation technique to the constant-weight fuzzy comprehensive evaluation methodology to see how effective it was in increasing the trust model’s performance. In this simulated experiment, rogue nodes were randomly placed over the network to conduct data-tampering attacks. We then investigated and contrasted the effects of the two approaches on nodes’ direct trust ratings in the same environment. 

Due to the data tampering attacks launched by malicious nodes, their data trust values tend to be relatively low. Therefore, in the variable-weight fuzzy synthesis evaluation algorithm, data trust indicators are assigned higher weights to highlight their impact on trust evaluation results. However, in the constant-weight algorithm, the weight of each trust indicator is set to a fixed value based on experience. This may lead to a situation where the final direct trust value remains high when other trust indicator values are high, failing to intuitively reflect the impact of defective trust indicators on trust evaluation results. As shown in Figure 3, when using the variable-weight algorithm, the trust of anomalous nodes rapidly decreases, demonstrating the algorithm’s rapid response to abnormal behavior. However, in the constant-weight algorithm, the presence of other trust indicators weakens the impact of defective trust indicators on trust evaluation results, resulting in a slow decline in trust values. Nevertheless, both algorithms perform well in reflecting the trust values of normal nodes. This result validates the effectiveness of the variable-weight algorithm in improving the performance of the trust model.

#### 5.1.2. Recommendation Filtering Mechanism

Bad-mouth attacks are a common method employed against trust recommendations. Malicious nodes modify the trust values of target nodes, disturbing the indirect trust assessment between normal and malicious nodes, jeopardizing network security. In this simulated experiment, we presume that hostile nodes simply supply bogus trust values and do not carry out other forms of attacks. We gradually increase the proportion of malicious nodes from 5% to 45% to observe their impact on the network trust mechanism. To comprehensively evaluate the effectiveness of the proposed anomaly node detection model, we compare the recommended trust values under three different scenarios. Firstly, we set an expected value, indicating the recommended trust value that target nodes should receive when there are no unreliable recommendation nodes in the network. Secondly, we apply the proposed recommendation filtering mechanism and calculate the filtered recommended trust values. Finally, we also consider a simple and direct method, averaging all recommended trust values without employing any filtering mechanism.

Figure 4 indicates that when the proportion of malicious nodes grows, the indirect trust ratings of regular nodes fall considerably under bad-mouth assaults. It is worth mentioning, however, that the proposed suggestion filtering strategy has only a little impact on indirect trust ratings. This is because, without a filtering mechanism, each node’s proposal value is treated equally, and all recommendation information in the network is accepted unconditionally. As the proportion of malicious nodes rises, so does the amount of incorrect information in the suggestion sequence. This has a significant impact on indirect trust scores. In contrast, the proposed filtering strategy is crucial for determining the appropriate trust levels. Using fuzzy closeness theory, this system successfully finds and filters out untrustworthy suggestion information, assuring the correctness and dependability of indirect trust ratings. This result clearly demonstrates the effectiveness of the proposed anomalous node detection strategy in countering bad-mouth assaults. 

### 5.2. Comparison to Other Works

In this simulated experiment, we compared it to two other trust models: ARTMM and LTrust. We identified the destination node as normal and ran simulations in which the connection quality varied between poor and excellent. 

Figure 5 indicates that low connection quality has a major influence on both the ARTMM and LTrust trust models. The proposed ATEM trust model, on the other hand, eliminates the effect of previous poor link quality on subsequent trust value evaluations by refusing to update trust values during times of low link quality. As a result, in complex underwater environments, ATEM can provide more accurate and consistent node trust values. 

To provide a more complete picture of ATEM’s performance, we ran a detection success rate test with a high malicious node rate of 30%. Under the influence of poor link quality, typical node trust values may fall below 0.5, and trust values from previous cycles may have an impact on the current cycle’s trust values. As the cumulative effect of low connection quality builds up, trust ratings continue to fall, potentially leading to more normal nodes being misclassified as malicious. However, as seen in Figure 6, the ATEM trust model successfully mitigates the fall in trust values of normal nodes by pausing trust value updates during periods of low connection quality. This reduces the possible influence on future trust value evaluations, considerably minimizing the chance of misidentifying normal nodes as malicious ones. As a result, in the ATEM trust model, there is a big disparity between the trust ratings of most normal nodes and malicious nodes, which leads to a significant increase in the success rate of harmful node identification. This finding indicates the ATEM trust model’s improved performance and stability in complicated network contexts.

## 6. Conclusions

This paper addresses two major challenges in UASNs: dynamic adjustment of trust indicator weights and filtering of recommended nodes. It proposes an adaptive trust evaluation model for detecting anomalous nodes in underwater acoustic sensor networks. The model aims to accurately calculate node trust values and effectively identify abnormal nodes in the network. In constructing the model, we comprehensively consider three categories of trust indicators and utilize a variable-weight fuzzy comprehensive evaluation algorithm for data fusion. By dynamically adjusting the weights of trust indicators, the model highlights the impact of defective trust indicators on trust evaluation, thereby improving the accuracy of direct trust evaluation. Subsequently, in the recommendation filtering strategy, the model based on fuzzy closeness theory effectively filters out recommended nodes with significant deviation and optimizes the calculation of indirect trust values based on deviation, reducing the influence of malicious recommended nodes on trust evaluation. Simulation results demonstrate that the model accurately identifies abnormal nodes, significantly enhancing the detection rate of the trust model.

In future research, we plan to further optimize the trust evaluation model, improve the link quality assessment method to more accurately reflect the actual link status, and consider the impact of environmental noise on underwater acoustic communication to further enhance network security and reliability.

## Figures and Tables

**Figure 1 sensors-24-02880-f001:**
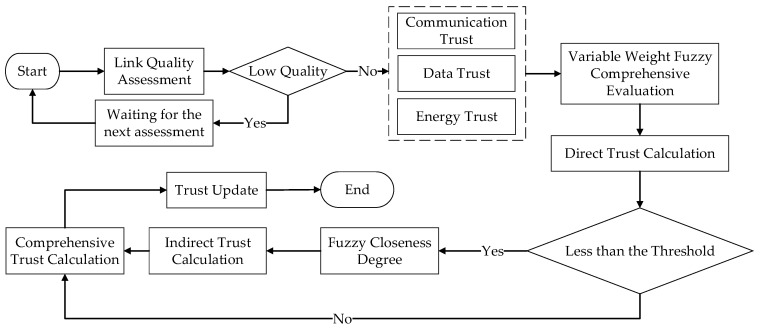
The structure of ATEM.

**Figure 2 sensors-24-02880-f002:**
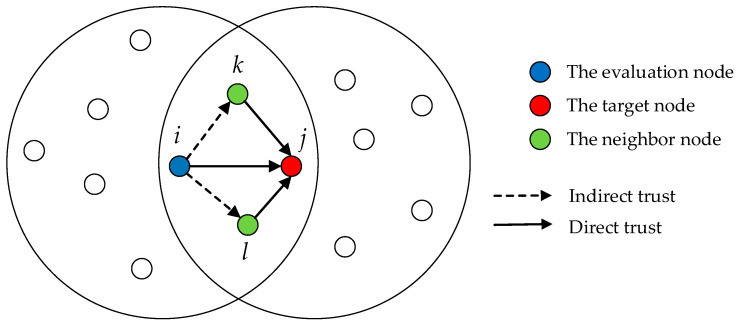
The relationship of indirect trust.

**Figure 3 sensors-24-02880-f003:**
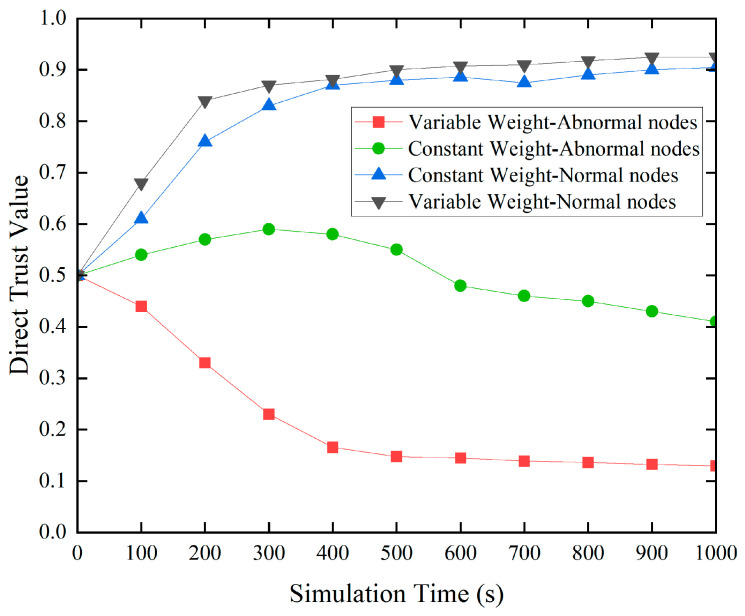
Variable weight and constant weight.

**Figure 4 sensors-24-02880-f004:**
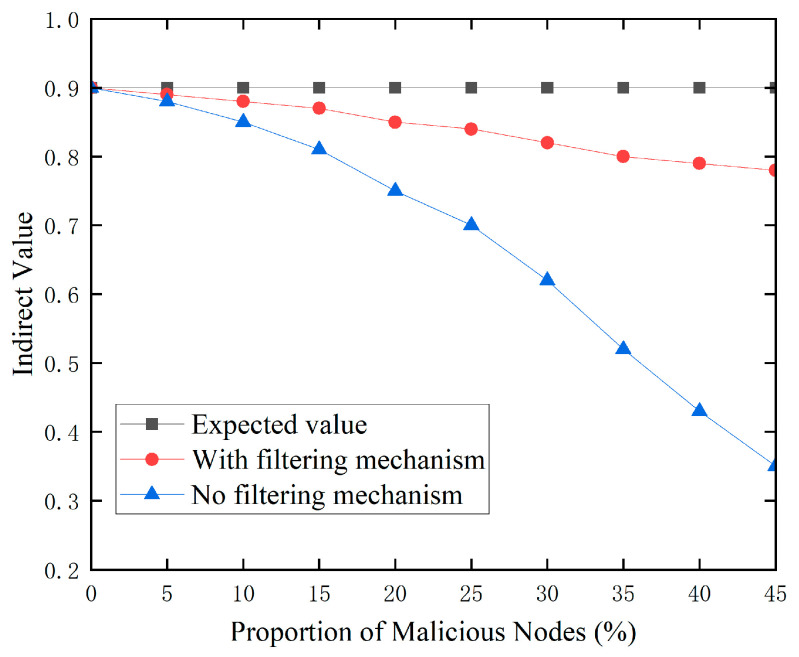
Recommendation filtering mechanism.

**Figure 5 sensors-24-02880-f005:**
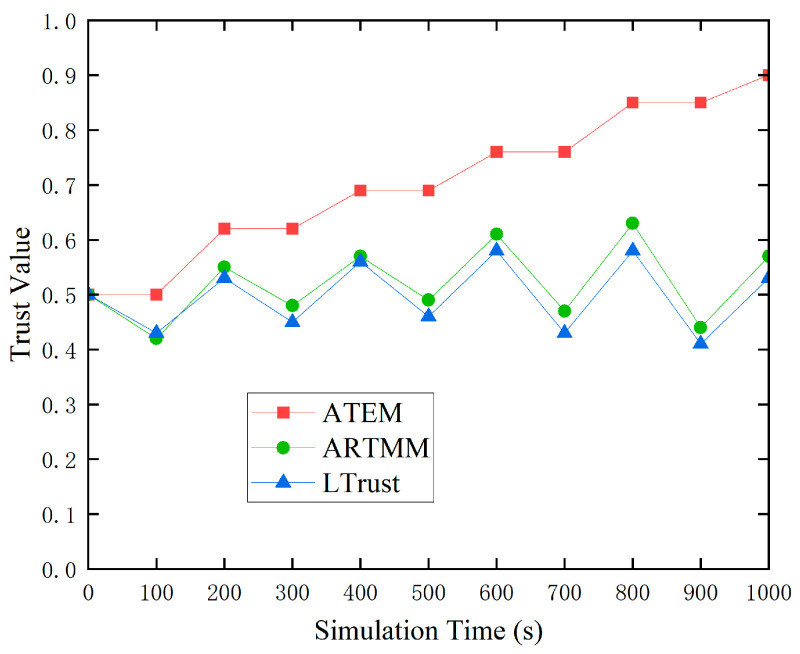
Dynamic variations in link quality.

**Figure 6 sensors-24-02880-f006:**
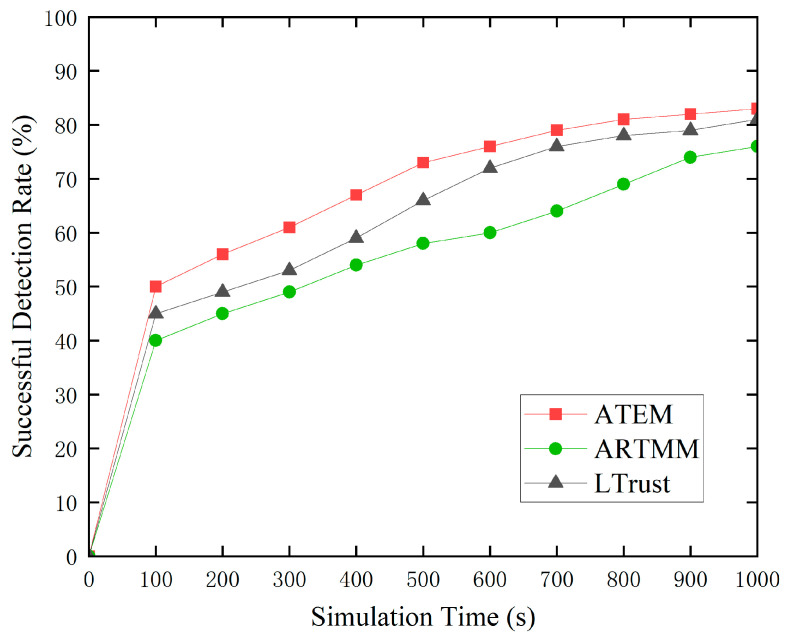
The proportion of malicious detections that are successful.

**Table 1 sensors-24-02880-t001:** Simulation parameters.

Parameters	Value
Network size	1000 m × 1000 m × 1000 m
Number of nodes	100
Node initial energy	1000 J
Communication radius	400 m
Node deployment method	Random

## Data Availability

Data are contained within the article.

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
