# Peer review of "An Adaptive Trust Evaluation Model for Detecting Abnormal Nodes in Underwater Acoustic Sensor Networks"

_sensors, 2024, doi:10.3390/s24092880_

Round 1
Reviewer 1 Report
Comments and Suggestions for Authors
Authors presented an Adaptive Trust Evaluation Model for Detecting Abnormal 2 Nodes in Underwater Acoustic Sensor Networks. The topic is interesting and aligned with the journal's scope. There are some aspects that need to be improved before accepting the paper. Following, I include a list of comments aimed at enhancing the quality of the paper:
1. The authors should avoid using acronyms in the abstract, especially if they are used only once. Check UASN and ATEM.
2. The introduction of the analyzed problem in the abstract is too long. Please reduce the provided information to just one or two sentences.
3. In the abstract, the authors have to highlight their results, including numerical values of the performance of their proposal.
4. It is recommended that at least 2 additional keywords be included.
5. The acronyms must be defined the first time they are used. Check UASN in the introduction, which is defined in the related work, and other possible cases.
6. There is a complete lack of references in the introduction, and it is not acceptable. The authors must add references to justify their affimrnations in the introduction and provide a better context for the readers. These are some references which must be added in the introduction given the topic of the paper
Underwater acoustic modems. IEEE Sensors Journal, 16(11), 4063-4071. (2015).
A survey on deployment techniques, localization algorithms, and research challenges for underwater acoustic sensor networks. International Journal of Communication Systems, 30(17), e3350. (2017).
7. At the beginning of a section which includes several subsections, please add a short paragraph detailing the content of the different subsections. Check Section 3 and other cases.
8. It is recommended that Section 4 be split into two sections. In the first section, the authors define the simulations that were conducted and the used metrics to assess the performance. The metrics can be moved from Section 3 to Section 4 to reduce the extension of Section 3. In the second section (new section 5), the authors must focus on the results.
9. The authors have to check the graphics and ensure that all of them have their axis' names and units correctly identified. Some encountered mistakes are: In Figure 3, the axis name "Time Period", units are missing or in Figure 4, the axis name "The proportion of malicious nodes" should be Proportion of Malicious Nodes (%)" or just "Malicious Nodes (%)"
10. At the end of the conclusions, the authors have to add the future work in an independent paragraph.
I do not find any major issue, just some typos.
Author Response
Dear Editor,
Thank you very much for your valuable comments. We have carefully studied your review comments and tried our best to modify the manuscript, explain some incomprehensible issues, and provide additional explanations for some of the issues pointed out.
The following are the point-by-point responses to the reviewers' comments:
Reviewer 1:
Comment 1: The authors should avoid using acronyms in the abstract, especially if they are used only once. Check UASN and ATEM.
Response: Thank you very much for your suggestions. We have carefully reviewed the abstract and removed the acronyms "UASN" and "ATEM" that were used only once to improve the readability and clarity of the abstract.
Comment 2: The introduction of the analyzed problem in the abstract is too long. Please reduce the provided information to just one or two sentences.
Response: Thank you very much for your guidance. We have streamlined the introduction of the problem in the abstract, compressing it into one or two sentences to highlight the core content and purpose of the study.
Comment 3: In the abstract, the authors have to highlight their results, including numerical values of the performance of their proposal.
Response: Thank you for your suggestion. We have highlighted our research results more clearly in the abstract, including specific performance values, so that readers can understand our research highlights more quickly.
Comment 4: It is recommended that at least 2 additional keywords be included.
Response: Thank you for your suggestion. We have added two related keywords based on the article content to improve the search visibility of the article and help readers better locate our research.
Comment 5: The acronyms must be defined the first time they are used. Check UASN in the introduction, which is defined in the related work, and other possible cases.
Response: Thank you for your careful correction. We have carefully reviewed the full text to ensure that all acronyms are clearly defined the first time they are used in the article to help readers better understand the technical terms used in the article.
Comment 6: There is a complete lack of references in the introduction, and it is not acceptable. The authors must add references to justify their affirmations in the introduction and provide a better context for the readers. These are some references which must be added in the introduction given the topic of the paper.
Underwater acoustic modems. IEEE Sensors Journal, 16(11), 4063-4071. (2015).
A survey on deployment techniques, localization algorithms, and research challenges for underwater acoustic sensor networks. International Journal of Communication Systems, 30(17), e3350. (2017).
Response: We apologize for the omission of not citing relevant literature in the introduction. Based on your suggestions, we have added relevant references to the introduction to support our arguments and provide readers with more detailed background information. Special thanks to you for recommending these two important documents, which we have included in the reference list.
Comment 7: At the beginning of a section which includes several subsections, please add a short paragraph detailing the content of the different subsections. Check Section 3 and other cases.
Response: Thank you for your suggestion. We have added a brief overview paragraph at the beginning of chapters containing subsections to help readers better understand the structure and organization of what they are about to read.
Comment 8: It is recommended that Section 4 be split into two sections. In the first section, the authors define the simulations that were conducted and the used metrics to assess the performance. The metrics can be moved from Section 3 to Section 4 to reduce the extension of Section 3. In the second section (new section 5), the authors must focus on the results.
Response: Your suggestions are very constructive. We have split the experimental part of Section 4 of the original article into two subsections according to your guidance, and split the network model and trust calculation of Section 3 of the original article into two independent parts. In the new structure, the content of the article is clearer and more hierarchical.
Comment 9: The authors have to check the graphics and ensure that all of them have their axis' names and units correctly identified. Some encountered mistakes are: In Figure 3, the axis name "Time Period", units are missing or in Figure 4, the axis name "The proportion of malicious nodes" should be Proportion of Malicious Nodes (%)" or just "Malicious Nodes (%)".
Response: Thank you very much for your careful review. The axis names and units of all charts have been reviewed and corrected based on your feedback. We ensure that all graphs and charts meet professional standards so that readers can accurately understand the data and information.
Comment 10: At the end of the conclusions, the authors have to add the future work in an independent paragraph.
Response: Thank you very much for reminding. We have added a separate paragraph at the end of the conclusion section dedicated to future work directions. Through such additions, we hope to provide readers with a more comprehensive research perspective and outlook.
Reviewer 2 Report
Comments and Suggestions for Authors
1. It is suggested to sort the literature out into different classifications, the main contributions of this paper should follow the literature review.
2. It is strongly recommended to add experimental validation aside from simulations.
3. Where do the input parameters come from? How could they represent the situation in reality?
4. Is there any other indicator to evaluate the outcomes, other than direct/indirect value? They’d better have real meanings in reality.
Comments on the Quality of English LanguageThe language quality is generally fine in this article.
Author Response
Dear Editor,
Thank you very much for your valuable comments. We have carefully studied your review comments and tried our best to modify the manuscript, explain some incomprehensible issues, and provide additional explanations for some of the issues pointed out.
The following are the point-by-point responses to the reviewers' comments:
Reviewer 2:
Comment 1: It is suggested to sort the literature out into different classifications, the main contributions of this paper should follow the literature review.
Response: Thank you very much for your valuable advice. We have reorganized the literature section and divided it into three categories based on the main contributions of this article so that readers can have a clearer understanding of the development status of each research field. At the same time, it better highlights our research highlights and innovations.
Comment 2: It is strongly recommended to add experimental validation aside from simulations.
Response: Thank you very much for your valuable opinion. I completely understand your concern about the importance of experimental verification in real-world environments. In this research field, experiments in real environments are often difficult to implement due to various practical constraints (such as cost, time, resources, etc.). Therefore, as you can see during the review, most studies in the existing literature also rely mainly on simulation experiments to verify the effectiveness of the scheme. Simulation experiments can provide us with valuable conclusions by conducting in-depth analysis and evaluation of complex systems while controlling variables. In our research, we use advanced simulation technology and rigorous experimental design to ensure the reliability and accuracy of simulation results. We believe that through these carefully designed simulation experiments, we can effectively verify the effectiveness of the proposed scheme and provide readers with convincing evidence. Of course, we also very much look forward to conducting further real-environment experiments when conditions permit in the future to more comprehensively verify our research results. Thank you again for your understanding and support, and we hope our explanation can eliminate your doubts.
Comment 3: Where do the input parameters come from? How could they represent the situation in reality?
Response: Thank you very much for your careful inquiry. Regarding the sources of input parameters, they are defined during the initialization phase of our simulation experiments. In order to ensure that these parameters can truly reflect the real situation, we refer to multiple relevant literatures and synthesize the data and suggestions therein to set these parameters, aiming to simulate the real environment as accurately as possible, thereby ensuring that our research has practical significance and Practical value. For example: in order to simulate the mobility of underwater nodes, we set the nodes to move according to the MCM model in the simulation experiment, hoping to simulate the underwater environment as much as possible. The MCM model is designed to capture the two main characteristics of water flow, namely currents and eddies. Typical paths alternate between fast downstream motion and cyclic motion, corresponding to sensor nodes in jets and vortices, respectively. We believe that our simulation experiments can be closer to the actual situation and provide strong support for the accuracy and reliability of the research results. Thank you again for your attention and we hope our explanation can answer your questions.
Comment 4: Is there any other indicator to evaluate the outcomes, other than direct/indirect value? They’d better have real meanings in reality.
Response: Thank you very much for your careful inquiry. Since the direct trust and indirect trust calculations in this article comprehensively consider a variety of trust indicators, such as communication, data, and energy, the performance of the model can be clearly reflected through changes in node trust values. In addition, in order to more comprehensively evaluate the effectiveness of this solution, this paper considers the two indicators of node comprehensive trust value and abnormal node successful detection rate when comparing it with the existing trust model when the link quality is unstable. These two indicators can further prove the superior performance and stability of this model in complex network environments. The comprehensive trust value of a node can reflect the credibility of the node in the real network and ensure the reliability of the data source. In addition, during the transmission process, nodes with high credibility are given priority for transmission, which can improve the security of the network. Thank you again for your valuable comments, I hope our explanation can answer your questions.
Round 2
Reviewer 1 Report
Comments and Suggestions for Authors
Authors have fixed all my comments. The paper is ready to be published.
Comments on the Quality of English LanguageI do not find English language errors.
Reviewer 2 Report
Comments and Suggestions for Authors
The questions aforementioned have been properly addressed.
Comments on the Quality of English LanguageEnglish is generally fine, minor grammer mistakes can be corrected.